# PsyAttention: Psychological Attention Model for Personality Detection

**Baohua Zhang, Yongyi Huang, Wenyao Cui, Huaping Zhang**[*] and **Jianyun Shang**

Beijing Institute of Technology, China

kevinzhang@bit.edu.cn

## Abstract

Work on personality detection has tended to incorporate psychological features from different personality models, such as BigFive and MBTI. There are more than 900 psychological features, each of which is helpful for personality detection. However, when used in combination, the application of different calculation standards among these features may result in interference between features calculated using distinct systems, thereby introducing noise and reducing performance. This paper adapts different psychological models in the proposed PsyAttention for personality detection, which can effectively encode psychological features, reducing their number by 85%. In experiments on the BigFive and MBTI models, PysAttention achieved average accuracy of 65.66% and 86.30%, respectively, outperforming state-of-the-art methods, indicating that it is effective at encoding psychological features.

## 1 Introduction

Personality detection helps people to manage themselves and understand themselves, and has been involved in job screening, personal assistants, recommendation systems, specialized health care, counseling psychotherapy, and political forecasting (Mehta et al., 2020b). Personality measures originated from psychology (Pervin and John, 1999; Matthews et al., 2003), and psychologists judge their patients' personality features through a series of tests.

The content that users post on social media can give valuable insights into their personalities (Christian et al., 2021). Personality detection can identify personality features from this, and often requires a combination of psychology, linguistics, and computer science.

There are many psychological models of individual personality, the most famous being the Big-Five (McCrae and Costa Jr, 1989; McCrae and John, 1992; Schmitt et al., 2008) and MBTI (Boyle, 1995; Hong, 2022; Celli and Lepri, 2018). The Big-Five model divides personality features into five factors(Digman, 1990): Openness (OPN), Conscientiousness (CON), Extraversion (EXT), Agreeableness (AGR), and Neuroticism (NEU). These are usually assessed through questionnaires in which people reflect on their typical patterns of thinking and behavior. The MBTI model describes personality by 16 types that result from the combination of binary categories in four dimensions[1]: (1) Extraversion (E) vs Introversion (I); (2) Thinking (T) vs Feeling (F); (3) Sensing (S) vs Intuition (N); and (4) Judging (J) vs Perceiving (P).

Previous work saw a person's use of language as a distillation of underlying drives, emotions, and thought patterns (Tausczik and Pennebaker, 2010; Boyd and Pennebaker, 2017). There are many theories and tools for extracting psychological features from texts, such as ARTE (Automatic Readability Tool For English) (Crossley et al., 2019), which has 55 psychological features); SEANCE (Sentiment Analysis And Cognition Engine) (Crossley et al., 2017), with 271 features; TAACO (Tool For The Automatic Analysis Of Cohesion) (Crossley et al., 2016), with 168 features; and TAALES (Tool For The Automatic Analysis Of Lexical Sophistication (Kyle and Crossley, 2015), with 491 features. These tools use multiple theories to obtain psychological features from text. For example, SEANCE uses eight theories, TAACO use fifteen theories. However, the application of different calculation standards among these features may result in interference between features calculated using distinct systems, thereby introducing noise and reducing performance. For example, when analyzing sentiment or emotion features, SEANCE uses more than 10 lexicons. A word's meaning may vary according to the lexicon, which leads to inconsistencies

---

[*]Corresponding author

[1]https://www.simplypsychology.org/the-myers-briggs-type-indicator.html

between features, and thus increases noise.

In the field of Natural Language Processing (NLP), personality detection aims to identify the personality features of individuals from their use of language. Work can be divided into two categories. One uses psychological features obtained by some statistical methods and uses machine learning tools such as Support Vector Machine (SVM) and K-Nearest Neighbor (KNN) for classification. Others take it as a text classification task, and directly input text to a neural network for classification. Work seldom combines psychological features with text embedding by treating them as sequences. However, because psychology and computer science are very different fields, psychologists often design features without considering how they will be introduced into a neural network model. NLP researchers ignore relations between those features, and focus on how to extract and encode more features, ignoring that this will introduce noise to the model, resulting in poor performance.

In this paper, we propose PsyAttention for personality detection, which can effectively encode psychological features and reduce their number by 85%. The main contributions of this paper are as follows.

- We propose PsyAttention to adapt different psychological models. Experimental results on two personality detection models demonstrate that the method can improve accuracy;

- We demonstrate that incorporating features designed by psychologists into neural networks is necessary, as pre-trained models fail to extract all of the psychological features present in the text;

- We select some important features in personality detection task. To the best of our knowledge, this is the first method to integrate, analyze, and reduce the number of psychological features for personality detection.

## 2 Related work

Research on individual personality detection focuses on how to extract more appropriate psychological features from texts and devise classification models. We introduce some methods below.

Methods to obtain personality features from text data generally use text analysis tools such as LIWC(Pennebaker et al., 2001) and NRC(Mohammad and Turney, 2013) to extract appropriate features, which are fed into standard machine learning classifiers such as SVM and Naïve Bayes (NB) (Mehta et al., 2020a) for classification. Abidin et al. (2020) used logistic regression (LR), KNN, and random forest, and found that random forest has the best performance. With the deeper study of psychology, language use has been linked to a wide range of psychological correlates (Park et al., 2015; Ireland and Pennebaker, 2010). Pennebaker and King (1999) proposed that writing style (e.g., frequency of word use) has correlations with personality. Khan et al. (2019) thought the social behavior of the user and grammatical information have a relation to the user's personality. It was found that combining common-sense knowledge with psycho-linguistic features resulted in a remarkable improvement in accuracy (Poria et al., 2013). Golbeck et al. (2011) proposed 74 psychological features; Nisha et al. (2022) extracted features like age, education level, nationality, first language, country, and religion, and found that XGBoost performed better than SVM and Naïve Bayes.

Methods based on deep-learning mainly regard personality detection as a classification task. Majumder et al. (2017) proposed to use CNN and Word2Vec embeddings for personality detection from text; since a document is too long, they proposed dividing the data into two parts and extracting the text features separately using two identical CNN models separately. Yu and Markov (2017) experimented with n-grams (extracted with a CNN) and bidirectional RNNs (forward and backward GRUs). Instead of using the pretrained Word2Vec model, they trained the word embedding matrix using the skip-gram method to incorporate internet slang, emoticons, and acronyms. Sun et al. (2018) used bidirectional LSTMs (BiLSTMs), concatenated with a CNN, and combined abstract features with posts to detect a user's personality. Ontoum and Chan (2022) proposed an RCNN model with three CNN layers and one BiLSTM layer. Xue et al. (2018) proposed an RCNN text encoder with an attention mechanism, inputting the text to a GRU and performing an attention calculation to make full use of the text features, then using GRU encodings to input the text in normal and reverse order. Similar to Xue et al. (2018), Sun et al. (2018) used an RNN and CNN, divided the text into segments, input them separately to LSTM, and input the merged re-

sults to the CNN to extract features. Some work has combined psychological features with deep learning models. Wiechmann et al. (2022) combined human reading behavior features with pretrained models. Kerz et al. (2022) used BiLSTM to encode 437 proposed features. Neuman et al. (2023) proposed an data augmentation methods for modeling human personality.

However, as the number of psychological features designed by psychologists increases, the models used to encode them are not improving, and remain a dense layer and BiLSTM. No research has focused on the effective use of psychological features. All models take them as a sequence, ignoring that they are obtained by a statistical approach. To better use psychological features, we propose Pysattention, along with a feature optimization method that can reduce the noise introduced by too many features.

## 3 Model

We introduce PsyAttention, whose model is shown in Figure 1, to effectively integrate text and psychological features. An attention-based encoder encodes numerical psychological features, and BERT is fine-tuned with psychological features obtained by tools to extract more personality information.

### 3.1 Psychological Feature Optimization

Kerz et al. (2022) proposed the most psychological features in the task of personality detection, and got them by automated text analysis systems such as CoCoGen(Marcus et al., 2016). To detect sentiment, emotion, and/or effect, they designed: (1) features of morpho-syntactic complexity; (2) features of lexical richness, diversity, and sophistication; (3) readability features; and (4) lexicon features. These groups of features can be divided into those of expressiveness and emotions contained in the text. To obtain more features, we choose three tools to analyze input texts: (1) SEANCE (Crossley et al., 2017) can obtain the features of sentiment, emotion, and/or effect; (2) TAACO (Crossley et al., 2016) can obtain features of cohesion; and (3) TAALES (Kyle and Crossley, 2015) can obtain features of expressive ability. Table1 shows the numbers of features used for each tool.

As mentioned above, too many features can introduce noise. Hence they should be used effectively. We calculate the Pearson coefficient between each feature and other features as a corre-

| Tools | Number of Features | Number of Features after Optimization | |
|---|---|---|---|
| | | MBTI | BigFive |
| SEANCE | 271 | 75 | 84 |
| TAACO | 168 | 6 | 9 |
| TAALES | 491 | 21 | 21 |
| TOTAL | 930 | 102 | 114 |

Table 1: Number of features used in model

Third and fourth columns: number of features used in MBTI and BigFive Essays datasets, respectively.

lation indicator. We set a threshold, and features with Pearson coefficients above this value are considered more relevant, and should be removed. To reduce the number of psychological features used, the feature with the largest correlation coefficients with other features was chosen to represent each group of correlated features. We leave fewer than 15% of those features, as shown in Table 1, and discussed in the Appendix.

### 3.2 Psychological Features Encoder

After we obtain and optimize numerical features, they must be encoded and merged with text embedding. How to encode these is a research challenge. Some researchers encode numerical sequences directly into the network by fully connected neural networks, but this is not effective because numeric sequences are too sparse for neural networks. Kerz et al. (2022) used BiLSTM as the encoder, but the numerical sequences are not temporally correlated. All of the psychological features are designed manually by psychologists. The 437 features used by Kerz et al. (2022) are discrete values and the order between them does not affect the result. In other words, feature extraction tools for psychological features, such as TAACO and TAALES, tend to prioritize the extraction of certain categories of features, such as emotional words followed by punctuation usage or others, when extracting features. These tools, which are designed artificially, do not take into account the order of the features, resulting in features that are disordered. Thus, we propose to encode them with a transformer based encoder. Since numerical sequences are not temporally correlated, we remove the position encoding. Taking $W = (w_0, w_1, w_2, \ldots, w_m)$ as the numerical features of psychology, and $F_{encoder}$ as the final output of the encoder, the process can be defined as follows:

$$H_{mutli\_att} = Mutil\_att(W) \quad (1)$$

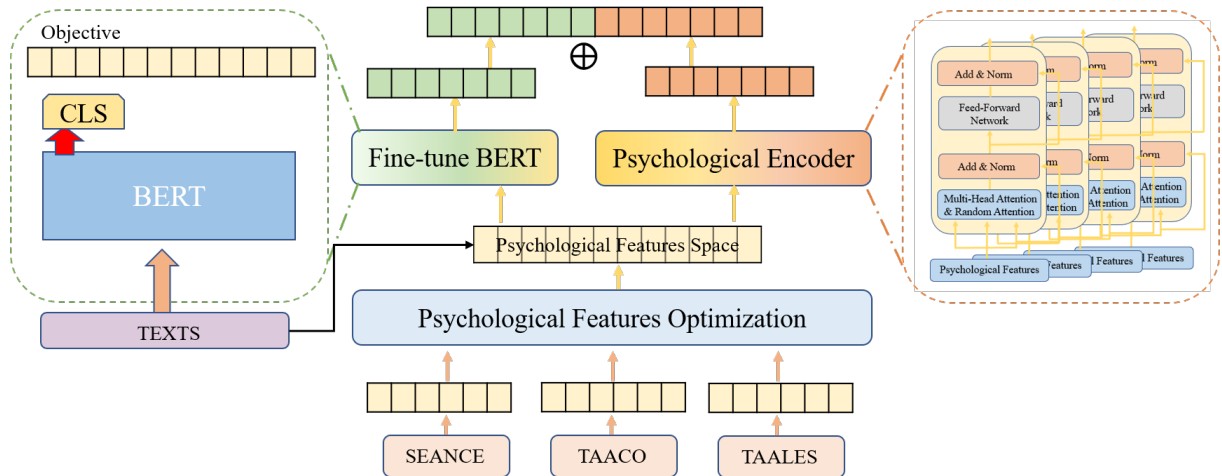

Figure 1: PsyAttention architecture. In Psychological Feature Selection, we select psychological features obtained from multiple tools by correlation analysis. In Psychological Encoder, an attention-based encoder encodes discrete numeric features. Fine-tuning BERT obtains more information about personality.

$$H_{LNorm1} = LayerNorm(H_{mutli\_att} + W) \quad (2)$$

$$H_{ffn} = FeedForward(H_{LNorm1}) \quad (3)$$

$$F_{encoder} = LayerNorm(H_{ffn} + H_{LNorm1}) \quad (4)$$

where $Mutil\_att$, $LayerNorm$, and $FeedForward$ denote multi-headed attention, the layer of Add & Norm, and the Feed-Forward Network, respectively.

### 3.3 Fine-tuning BERT

We use BERT (Kenton and Toutanova, 2019) for text feature extraction. The pretrained model has proven its superior performance in most natural language processing tasks. Taking $T = (x_0, \cdot, x_m)$ as the input text, we only use [CLS] as the text embedding. As [CLS] is the first token of the BERT output, the process can be defined as $H = h_1 \cdots h_m = BERT(x_1 \cdots x_n)[0]$.

Then, we employ a dense layer to obtain a vector that has the same dimensional as the psychological feature $p_1 \cdots p_l$ .

$$f_1 \cdots f_l = Dense(H) \quad (5)$$

The loss function is shown in formula 6. We fine-tune BERT to make the vector of CLS more similar to the psychological feature vector.

$$loss = 1 - cos\_sim(\{f_1 \cdots f_l\}, \{p_1 \cdots p_l\}) \quad (6)$$

It is noteworthy that during the training process, we first conduct training for this step. After fine-tuning BERT, we then fix the current weights and use them to extract textual features.

### 3.4 Personality detection Using PsyAttention model

We design an embedding fusion layer to control the information that can be used in the final representation. We use a dense layer to calculate the weight, taking $F_{PSY}$ and $F_{BERT}$ to indicate the embedding of the psychological encoder and BERT, respectively. We can obtain the weights as follow formulas.

$$\begin{aligned} W_{PSY} &= Dense(F_{PSY}) \\ W_{BERT} &= Dense(F_{BERT}) \end{aligned} \quad (7)$$

Due to the fact that the extractable psychological and textual features are determined by the original text, we employ a dynamic weighting scheme to integrate the psychological and textual features. We can obtain the final embedding by following formula.

$$Final_{emb} = F_{PSY} \cdot W_{PSY} | F_{BERT} \cdot W_{BERT} \quad (8)$$

where $(\cdot \mid \cdot)$ is the concatenation operator.

After we get the final representation, we input them to a fully connected neural network for classification. Taking $P$ as the final result, the process can be defined as

$$P = Classifier(Final_{emb}) \quad (9)$$

It is worth noting that the final loss function of our model is the cross-entropy, where formula (6) is the loss function for the Fine-tune Bert component only.

|       | O    | C    | E    | A    | N    |
|-------|------|------|------|------|------|
| Ture  | 1272 | 1254 | 1277 | 1310 | 1233 |
| False | 1196 | 1214 | 1191 | 1158 | 1235 |
| Total | 2468 | 2468 | 2468 | 2468 | 2468 |

Table 2: The statistics of BigFive Essays dataset.

| MBTI Kaggle        | I/E  | N/S  | T/F  | P/J  |
|--------------------|------|------|------|------|
| I or N or T or P   | 6676 | 7478 | 3981 | 3434 |
| E or S or F or J   | 1999 | 1197 | 4694 | 5241 |
| Total              | 8675 | 8675 | 8675 | 8675 |

Table 3: The statistics of MBTI Kaggle dataset.

# 4 Experiments

## 4.1 Dataset

We conducted experiments on two widely used personality benchmark datasets: BigFive Essay dataset(Pennebaker and King, 1999) and MBTI Kaggle dataset (Li et al., 2018). The BigFive Essay dataset consists of 2468 essays written by students and annotated with binary labels of the BigFive personality features, which were obtained through a standardized self-reporting questionnaire. The average text length is 672 words, and the dataset contains approximately 1.6 million words. The MBTI dataset consists of samples of social media interactions from 8675 users, all of whom have indicated their MBTI type. The average text length is 1,288 words. The dataset consists of approximately 11.2 million words. The detailed statistics are shown in Table 2 and Table 3.

To retain the psychological features contained in the text, we perform no pre-processing on the text when extracting psychological features using text analysis tools, and we remove noise such as concatenation, multilingual punctuation, and multiple spaces before input to the BERT model.

## 4.2 Parameter Settings

We utilized Python 3.8.1, PyTorch 1.13.0, Transformers 4.24.0, and scikit-learn 1.1.3 for implementing our model. Our training process involved 4 NVIDIA RTX 3090 GPUs. We trained our model with 50 epochs. We have employed a split ratio of 7:2:1 for training, validation, and testing respectively.

We researched the number of psychological encoders that can obtain the best result, used even numbers from 2 to 12 to test the results, and found that the model obtained the best result with 8 en-

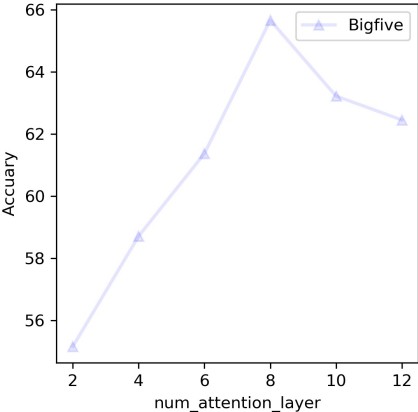

Figure 2: Results of different numbers of attention layers on BigFive dataset

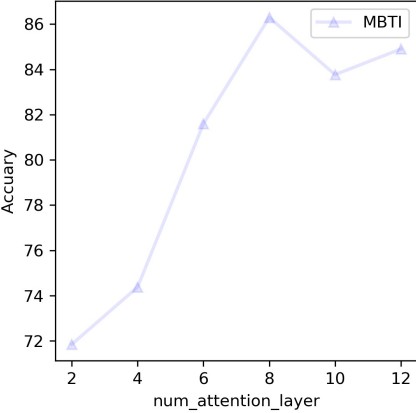

Figure 3: The results of different numbers of attention layers on MBTI dataset

coders. The results are shown in Figures 2 and 3.

Taking BERT-base as the text encoder model, we used four attention head layers, eight psychological feature encoder layers, a learning rate of 2e-5, and the Adam optimizer. The maximum text length was 510, the length of psychological features was 108, and other parameters were the same as for BERT-base.

## 4.3 Baselines and Results

We took the BERT+PSYLING Ensemble model of Kerz et al. (2022) as our baseline , who used 437 text contours of psycholinguistic features, with BERT as the text encoder. While we selected 108 psychological features and designed a psychological feature encoder based on an attention module, which we think is more effective. We also com-

| | BigFive Essays | | | | | | MBTI Kaggle | | | | |
|---|---|---|---|---|---|---|---|---|---|---|---|
| | O | C | E | A | N | Avg | I/E | N/S | T/F | P/J | Avg |
| Majumder et al. (2017) | 61.1 | 56.7 | 58.1 | 56.7 | 57.3 | 58 | - | - | - | - | - |
| Kazameini et al. (2020) | 62.1 | 57.8 | 59.3 | 56.5 | 59.4 | 59 | - | - | - | - | - |
| Amirhosseini (2020) | - | - | - | - | - | - | 79 | 86 | 74.2 | 65.4 | 76.1 |
| psychological+MLP | 60.4 | 57.3 | 56.9 | 57 | 59.8 | 58.3 | 77.6 | 86.3 | 72 | 61.9 | 74.5 |
| BERT-base+MLP | 64.6 | 59.2 | 60 | 58.8 | 60.5 | 60.6 | 78.3 | 86.4 | 74.4 | 64.4 | 75.9 |
| All features+MLP | 61.1 | 57.4 | 57.9 | 58.6 | 60.5 | 59.1 | 78.4 | 86.6 | 75.9 | 64.4 | 76.3 |
| BERT-large+MLP | 63.4 | 58.9 | 59.2 | 58.3 | 58.9 | 59.7 | 78.8 | 86.3 | 76.1 | 67.2 | 77.1 |
| Ramezani et al. (2022) | 56.30 | 59.18 | 64.25 | 60.31 | 61.14 | 60.24 | - | - | - | - | - |
| Kerz et al. (2022)(single) | 66.23 | 60.60 | 61.61 | 61.05 | 61.65 | 62.28 | 86.25 | 90.96 | 84.66 | 79.65 | 85.37 |
| BERT | 56.67 | 57.35 | 57.22 | 58.91 | 59.40 | 57.91 | 77.74 | 86.27 | 60.09 | 54.67 | 69.69 |
| BERT+Dense | 61.83 | 56.91 | 57.95 | 60.55 | 57.96 | 59.04 | 83.97 | 84.76 | 80.95 | 76.0 | 81.42 |
| BERT+BiLSTM | 62.23 | 58.94 | 58.42 | 59.16 | 59.37 | 59.62 | 84.28 | 88.79 | 83.95 | 76.57 | 83.40 |
| PsyAttention | **68.62** | **64.21** | **64.43** | **66.75** | **64.27** | **65.66** | **87.94** | **91.47** | **85.24** | **80.53** | **86.30** |

Table 4: Results (classification accuracy) of Personality Detection

pared our model with other models. Ramezani et al. (2022) proposed a Hierarchical Attention Network (HAN) combined with base method predictions to carry out stacking in document-level classification. They combined five basic model predictions with the text features, and input them to a dense layer to get the final prediction. Mehta et al. (2020a) also used psychological features with BERT, and experimented with logistic regression, SVM, a multilayer perceptron (MLP) with 50 hidden units, and machine learning models such as those of Amirhosseini (2020), Majumder et al. (2017), and Kazameini et al. (2020).

We conducted experiments on the BigFive Essays and Kaggle MBTI Kaggle datasets. The results are shown in Table 4. The PsyAttention listed in Table 4 is our model. Majumder et al. (2017),Kazameini et al. (2020) and Amirhosseini (2020) represent the results of the machine learning models. The others represent the results of the deep learning models. As we can see, the average accuracy of machine learning is lower than that of the deep learning model, which indicated that machine learing models are not as effective as deep learning models. Kerz et al. (2022)(single) is the model that named BERT+ATTN-PSYLING (FT) in Kerz et al. (2022)'s work. Since we did not employ the stacking approach in our model, we only compared our model with Kerz et al. (2022)(single). Both PsyAttention and Kerz et al. (2022)(single) use deep learning models and psychological features. As we can see from Table 4, the accuracy of PsyAttention is 2% higher than Kerz et al. (2022)(single) on the BigFive datasets and 0.3% higher on the MBTI dataset, which proves that our model is better than Kerz et al. (2022)(single). The

architecture of BERT+BiLSTM is same as that of the of Kerz et al. (2022) (single), the difference is that we use all 930 psychological features while Kerz et al. (2022) (single) only use 437. The accuracy of BERT+BiLSTM is lower than that of Kerz et al. (2022)(single), which proves that more features will cause more noise.

## 4.4 Ablation Experiments

We performed ablation experiments, and the results of which are shown in Table 5. '-psyencoder' indicates that we do not use the Psychological Encoder, '-BERT' indicates that we do not use the Fine-tune BERT, '-weight' indicates that we do not use the embedding fusion weight, '$-BERT_{psy}$' indicates that we do not use psychological features in Fine-tune BERT, and '-Features_Selection ' indicates that we use all 930 psychological features. As we can see, the results for '-BERT' are reduced the most, indicating that the BERT coding component is the most important, and the results for '-psyencoder' are also reduced, indicating that psychological features also have a significant effect on the results. The results of '-Features_Selection ' indicate that there is some noise in all features, and it is right to do the feature selection.

## 4.5 Psychological Features In Pretrained Models

We designed a simple experiment to verify whether text embedding includes psychological information. Consider that almost every model in NLP tasks takes word embedding as input and uses some neural networks to extract features, which are input to a discriminant model or generative model to get the final representation. If there is psychological information in the text embedding, we can extract

|  | BigFive Essays | | | | | | MBTI Kaggle | | | | |
|---|---|---|---|---|---|---|---|---|---|---|---|
|  | O | C | E | A | N | Avg | I/E | N/S | T/F | P/J | Avg |
| PsyAttention | **68.62** | **64.21** | **64.43** | **66.75** | **64.27** | **65.66** | **87.94** | **91.47** | **85.24** | **80.53** | **86.30** |
| -psyencoder | 61.69 | 58.13 | 58.06 | 60.59 | 58.19 | 59.33 | 84.01 | 86.94 | 82.30 | 77.72 | 82.74 |
| -BERT | 59.92 | 58.05 | 55.94 | 58.63 | 56.79 | 57.87 | 78.78 | 83.12 | 75.98 | 77.53 | 78.85 |
| -weight | 66.63 | 62.89 | 62.20 | 65.55 | 62.90 | 64.03 | 86.61 | 88.26 | 82.19 | 77.75 | 83.70 |
| -BERT$_{psy}$ | 64.58 | 61.19 | 60.54 | 64.24 | 62.48 | 62.63 | 83.02 | 86.55 | 80.59 | 74.94 | 81.28 |
| -Features_Selection | 65.98 | 62.18 | 63.32 | 64.13 | 60.76 | 63.27 | 85.52 | 91.03 | 84.35 | 78.98 | 84.97 |

Table 5: Ablation experiment results (classification accuracy)

| Methods | MBTI | BigFive |
|---|---|---|
| BERT | 0.575 | 0.551 |
| BERT & Fine-tune | 0.687 | 0.662 |

Table 6: Results of cosine similarity

it by some networks. Hence we designed a model to extract psychological features from text embedding, used BERT to obtain the text embedding, and a dense layer to extract psychological features. The objective was to make the vector of CLS more similar to the psychological feature vector. We calculated the cosine similarity between the features extracted from the text embedding and those calculated by the psychology tool, and took it as the loss function and evaluation metric of the results. The experimental results are shown in Table 6.

We used a BERT-base model. Table 6 shows the results of two encoders, where "BERT" indicates the direct use of BERT-base to obtain the text embedding, without fine-tuning on psychological datasets; BERT&Fine-tune is obtained by fine-tuning BERT-base on those two datasets. During training, we fixed the weight of BERT&Fine-tune but did not fix the weight of BERT. As we can see, even if we fine-tune BERT-base, we can only get a cosine similarity of less than 0.7, which indicates that there is little psychological information in text embedding, and we need to introduce it in the model.

### 4.6 The Weights of Psychological Features

After removing a substantial number of features, we sought to investigate the remaining features' impact on the psychological discrimination task. To accomplish this, we introduced an attention mechanism to calculate the attention scores of the psychological features and explore the importance of different features. Because psychological features represented by MBTI and BigFive differ, we separately calculated the attention scores for their

corresponding psychological features. After normalization, we plotted the feature distribution of the psychological features contained in both MBTI and BigFive, showed in Figure 4.

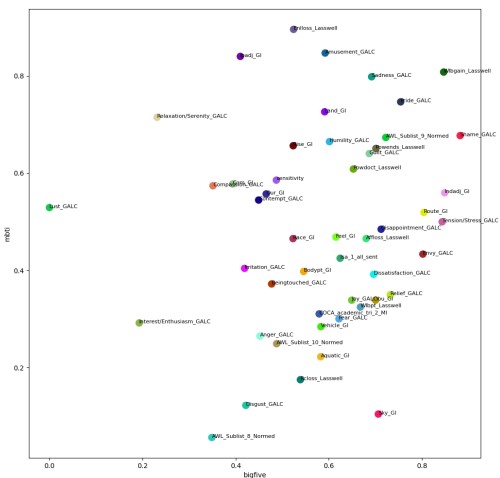

Figure 4: The common psychological feature distribution used in the MBTI and BigFive Essays datasets, where the x-axis represents the attention score on Big-Five and the y-axis represents MBTI, with larger values indicating more important features. We will introduce more in section 6.2.

Table 7 displays the features that scored above 0.6 after separate normalization on the two datasets. "AWL_Sublist_9_Normed" indicates the number of words in the text that belong to the Academic Word List Sublist 9. "Powends_Lasswell" and "Powdoct_Lasswell" both represent words related to dominance, respect, money, and power. The former includes words such as "ambition," "ambitious," "arrangement," "bring," and "challenge," while the latter includes words like "bourgeois," "bourgeoisie," "capitalism," "civil," and "collectivism."

It can be observed that the aforementioned features can be categorized into three groups:

Table 7: The more important psychological features

| Features | MBTI | BigFive | Avg | Variable description |
|---|---|---|---|---|
| Wlbgain_Lasswell | 0.85 | 0.81 | 0.83 | Well being gain: 37 various words related to a gain in well being. |
| Shame_GALC | 0.88 | 0.68 | 0.78 | Shame |
| Sadness_GALC | 0.69 | 0.80 | 0.75 | Sadness |
| Pride_GALC | 0.75 | 0.75 | 0.75 | Pride |
| AWL_Sublist_9_Normed | 0.72 | 0.67 | 0.70 | Academic Word List Sublist 9 |
| Powends_Lasswell | 0.70 | 0.65 | 0.68 | Power End: 30 words about the goals of the power process. |
| Guilt_GALC | 0.69 | 0.64 | 0.66 | Guilt |
| Humility_GALC | 0.60 | 0.66 | 0.63 | Humility |
| Powdoct_Lasswell | 0.65 | 0.61 | 0.63 | Power doctrine: 42 words for recognized ideas about power relations and practices. |

personal emotion-related, personal realization-related, and word usage habit-related. The primary personal emotion-related features are Wlbgain_Lasswell, Shame_GALC, Sadness_GALC, Pride_GALC, Guilt_GALC, and Humility_GALC, which encompass both positive and negative emotions experienced by individuals. The personal realization-related features mainly consist of Powends_Lasswell and Powdoct_Lasswell, which are indicative of self-actualization and personal development. Lastly, the word usage habit feature, AWL_Sublist_9_Normed, relates to the frequency of academic words used by individuals and is believed to reflect their level of rigor. The proportion of words in the personal emotion category is over 66%, indicating that emotion is a critical characteristic in personal character discrimination. The proportion of words in the self-realization category is over 20%, reflecting the importance of self-actualization in personal character. The word usage habit feature is believed to be mainly related to whether an individual is rigorous or not, as casual individuals tend to use fewer academic words.

It should be noted that not all emotions are included in the personal emotion category, such as Loneliness, Envy, and Anxiety. This may be due to the fact that these emotions are often generated during interpersonal interactions, and therefore may not be as critical for personality detection. Furthermore, it should be noted that emotions represent only a portion of the content, and other self-actualization-related characteristics may be even more significant. Personal realization-related features, such as Powends_Lasswell and Powdoct_Lasswell, are indicative of an individual's self-actualization and personal development, which are critical components of personal character.

## 5 Conclusion

We proposed PysAttention for personality detection, which can effectively encode psychological features, and reduce their number by 85%. We categorized the psychological features contained in the text as either expressiveness or emotions, and extracted the relative features using three text analysis tools. We demonstrated that the pre-trained model contained only a small amount of psychological information, and proposed a method of psychological feature selection. Unlike models that simply encode a long series of numerical psychological features through BiLSTM, we efficiently encode psychological features with a transformer-like encoder. We also investigate the impact of different psychological features on the results and employ an attention mechanism to identify the more significant features. In experiments on two classical psychological assessment datasets, the proposed model outperformed state-of-the-art algorithms in terms of accuracy, demonstrating the effectiveness of PysAttention.

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

## Limitations

This paper has the following limitations: 1) The BERT encoder cannot completely encode text sequences with a length greater than 512, but almost every sentence in the BigFive dataset is longer than that. We will consider models such as long-former, but this also cannot deal with longer sentences; 2) The research on psychological features is still limited, and there may be some important ones we did not include in our model.

## Ethics Statement

All work in this paper adheres to the ACL Ethics Policy.

## Acknowledgements

This work was supported by Fundamental Strengthening Program Technology Field Fund, China (2022-JCJQ-JJ-0930).

## 6 Appendix

### 6.1 The correlation coefficients for all psychological features

We introduce the psychological feature optimization method. We calculated correlation coefficients for all psychological features on both datasets, and plotted the heatmaps shown in Figure 7. The many light-colored blocks in the diagram indicate a large number of highly relevant features. We grouped features with correlation coefficients greater than 0.2, selected those with the highest number of correlated features from each group as representative,

and did not use the rest. Table 8 shows some features with correlation coefficients; after processing feature selection, we only use Joy and Disgust.

We show the features that we used in Table 9 and Table 10. Figure 6 shows the heatmaps of correlation coefficients for the features we used in two datasets. There are no light-colored blocks, which indicates that these features have low relevance.

## 6.2 The Attention Score of Features

Figure 7 is the importance score on the BigFive and MBTI Kaggle datasets after summing and normalizing the attention scores obtained by inferring all the test data. Figure 8 is the bigger version of Figure 4.

The objective of this study is to investigate the relative importance of different features using attention scores. By analyzing the attention scores assigned to each feature, we aim to identify the most influential characteristics in personality detection.

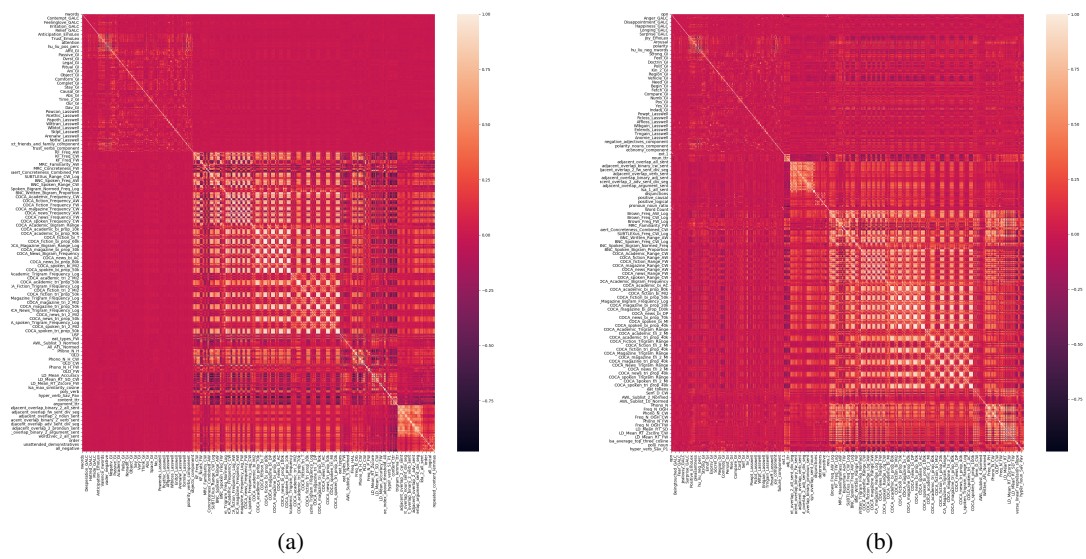

Figure 5: Correlation coefficients for all psychological features on both datasets. (a) MBTI; (b) BigFive.

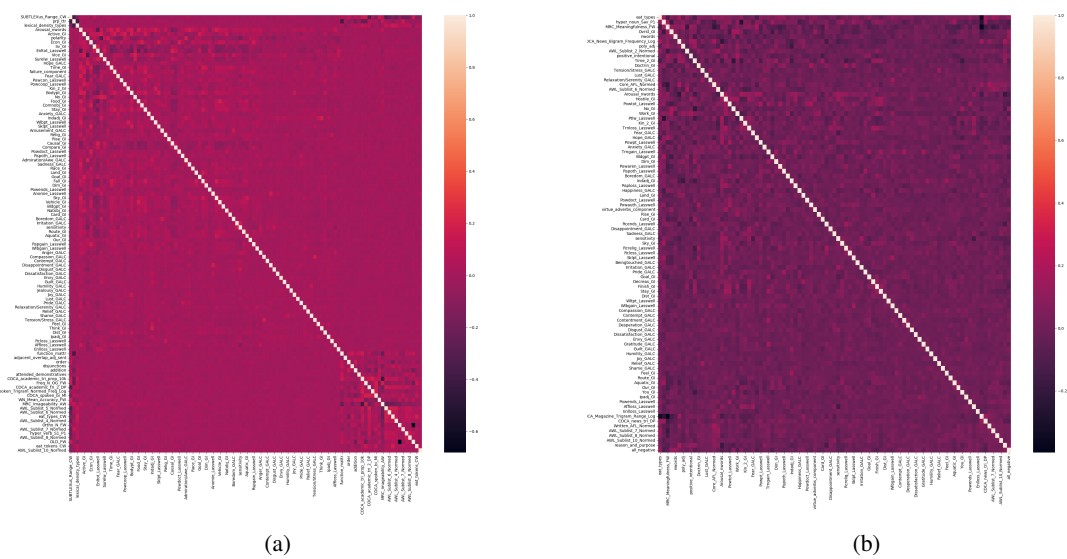

Figure 6: Correlation coefficients for psychological features after selection on both datasets. (a) MBTI; (b): BigFive.

| Feelinglove | Disgust | Fear | Joy | Negative | Positive | Sadness | Surprise | Trust |
|---|---|---|---|---|---|---|---|---|
| 1 | -0.02 | -0.01 | 0.46 | -0.08 | 0.29 | 0.01 | 0.03 | 0.28 |
| -0.02 | 1 | 0.55 | 0.08 | 0.73 | 0.02 | 0.63 | 0.15 | 0.04 |
| -0.01 | 0.55 | 1 | 0.1 | 0.65 | 0.1 | 0.68 | 0.19 | 0.15 |
| 0.46 | 0.08 | 0.1 | 1 | -0.04 | 0.77 | 0.09 | 0.5 | 0.71 |
| -0.08 | 0.73 | 0.65 | -0.04 | 1 | -0.05 | 0.73 | 0.14 | -0.04 |
| 0.29 | 0.02 | 0.1 | 0.77 | -0.05 | 1 | 0.04 | 0.36 | 0.73 |
| 0.01 | 0.63 | 0.68 | 0.09 | 0.73 | 0.04 | 1 | 0.19 | 0.05 |
| 0.03 | 0.15 | 0.19 | 0.5 | 0.14 | 0.36 | 0.19 | 1 | 0.4 |
| 0.28 | 0.04 | 0.15 | 0.71 | -0.04 | 0.73 | 0.05 | 0.4 | 1 |

Table 8: Example for feature selection

| | | | | | |
|---|---|---|---|---|---|
| Arousal_nwords | Comnobj_GI | Fall_GI | Compassion_GALC | Dist_GI | Freq_N_OG_FW |
| Active_GI | Stay_GI | Dim_GI | Contempt_GALC | Ipadj_GI | COCA_academic_tri_2_DP |
| polarity | Anxiety_GALC | Powends_Lasswell | Disappointment_GALC | Rcloss_Lasswell | BNC_Spoken_Trigram_Normed_Freq_Log |
| Econ_GI | Indadj_GI | Anomie_Lasswell | Disgust_GALC | Affloss_Lasswell | COCA_spoken_bi_MI |
| Sv_GI | Wlbpt_Lasswell | Sky_GI | Dissatisfaction_GALC | Enlloss_Lasswell | WN_Mean_Accuracy_CW |
| Enltot_Lasswell | Sklpt_Lasswell | Vehicle_GI | Envy_GALC | lsa_1_all_sent | WN_Mean_Accuracy_FW |
| Vice_GI | Amusement_GALC | Bldgpt_GI | Guilt_GALC | function_mattr | MRC_Imageability_AW |
| Surelw_Lasswell | Relig_GI | Natobj_GI | Humility_GALC | adjacent_overlap_adj_sent | AWL_Sublist_5_Normed |
| Hope_GALC | Rise_GI | Card_GI | Jealousy_GALC | prp_ttr | AWL_Sublist_6_Normed |
| Time_GI | Causal_GI | Boredom_GALC | Joy_GALC | lexical_density_types | eat_types_CW |

Table 9: Feature names used in MBTI dataset

| | | | | | |
|---|---|---|---|---|---|
| Arousal_nwords | Powpt_Lasswell | Rise_GI | Wltpt_Lasswell | Route_GI | AWL_Sublist_5_Normed |
| Valence | Anxiety_GALC | Card_GI | Wlbgain_Lasswell | Aquatic_GI | AWL_Sublist_3_Normed |
| Hostile_GI | Trngain_Lasswell | Rcends_Lasswell | nwords | Our_GI | AWL_Sublist_6_Normed |
| Posaff_Lasswell | Bldgpt_GI | Disappointment_GALC | Compassion_GALC | You_GI | AWL_Sublist_7_Normed |
| Passive_GI | Dim_GI | Sadness_GALC | Contempt_GALC | Ipadj_GI | AWL_Sublist_8_Normed |
| Powtot_Lasswell | Powaren_Lasswell | sensitivity | Contentment_GALC | Powends_Lasswell | Word Count |
| Ovrst_GI | Rspoth_Lasswell | Sky_GI | Desperation_GALC | Affloss_Lasswell | AWL_Sublist_4_Normed |
| Time_2_GI | Boredom_GALC | Rcrelig_Lasswell | Disgust_GALC | Enlloss_Lasswell | AWL_Sublist_9_Normed |
| Econ_GI | Indadj_GI | Rcloss_Lasswell | Dissatisfaction_GALC | COCA_News_Bigram_Frequency_Log | AWL_Sublist_10_Normed |
| Enltot_Lasswell | Rsploss_Lasswell | Sklpt_Lasswell | Envy_GALC | MRC_Meaningfulness_FW | lsa_1_all_sent |

Table 10: Feature names used in BigFive dataset

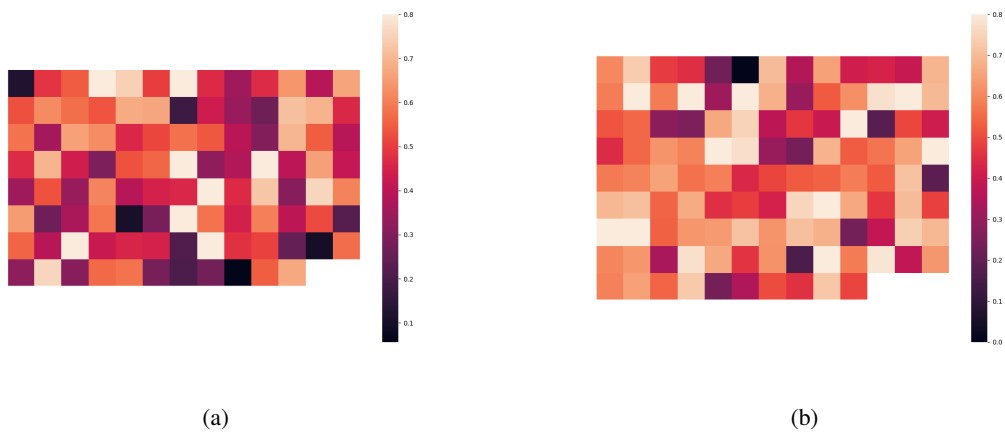

(a)  (b)

Figure 7: The attention score for all psychological features on both datasets. (a) MBTI; (b) BigFive.

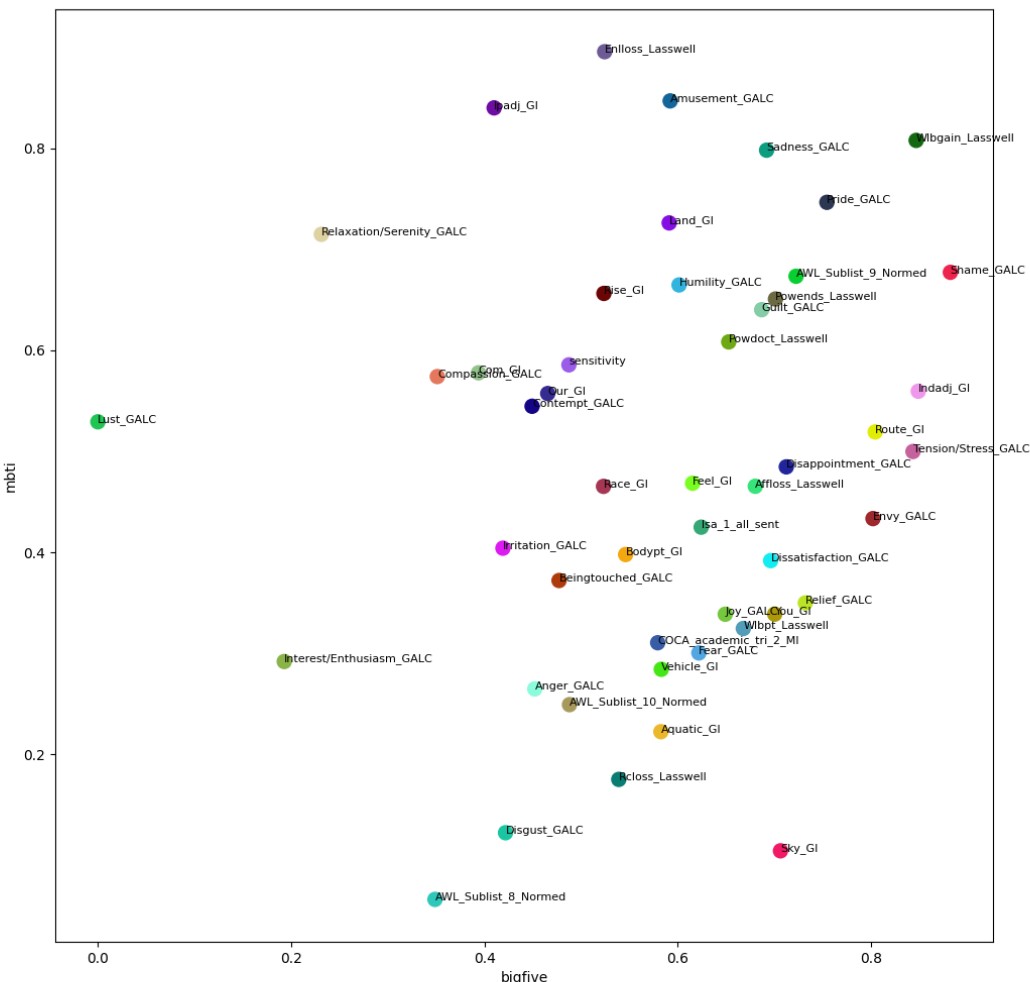

Figure 8: The common psychological feature distribution used in the MBTI Kaggle dataset and BigFive Essays dataset, where the x-axis represents the attention score on BigFive and the y-axis represents MBTI, with larger values indicating more important features.

