# OpenReview forum: "PsyAttention: Psychological Attention Model for Personality Detection"
_EMNLP/2023/Conference — EMNLP 2023 Findings_

### Official Review · Reviewer_QLop · 2023-07-25

**Soundness:** 3

**Excitement:**

3: Ambivalent: It has merits (e.g., it reports state-of-the-art results, the idea is nice), but there are key weaknesses (e.g., it describes incremental work), and it can significantly benefit from another round of revision. However, I won't object to accepting it if my co-reviewers champion it.

**Paper Topic And Main Contributions:**

The paper presents the results of an *NLP engineering experiment*. On the one hand it describes PsyAttention as an approach to predict dimensions (e.g., from BigFive or MBTI) of personality of an author of a text using an Psychological Encoder and and a fine-tuned BERT model.

One contribution is the optimization of the number of features that represent personality in texts. Another contribution is the implementation of a model that consists out of a fine-tuned BERT model and a "transformer like" model that uses an attention mechanism to process feature values which are extracted from text.

**Questions For The Authors:**

Question A: Which attention mechanism was used for the psychological feature encoder?
Question B: How can any attention mechanism work on a sequence of discrete numerical values that do not represent meanings/semantics but just arbitrary values?
Question C: How are the features represented when used as input for the encoder (this perhaps answers Question B too)?
Question D: Can you describe how the datasets have been used for the evaluation (e.g., train/test/validation split, cross-validation (yes/no), distribution of labels in the dataset)?
Question E: As the balance of labels in the datasets not clear, what is the F1 measure of your results?
Question F: Can motivate why it is useful to compare the text and feature embeddings by the cosine similarity?

**Reasons To Accept:**

- The paper proposes an interesting approach to predict the personality of an author of a text by processing that text.
- The authors provide a good overview on methods to extract personality-related (linguistic) features from text

**Reasons To Reject:**

Missing reproducibility:
- It is not described or motivated how the attention mechanism on sequences of numerical values works
- The evaluation experiment is not described (e.g, basic like train/test/validation split ratio or number of folds in case of a cross-validation are missing)
- The input-representation of the discrete features values to the psychological feature encoder is not described (or at least shown as an example)

**Reproducibility:**

3: Could reproduce the results with some difficulty. The settings of parameters are underspecified or subjectively determined; the training/evaluation data are not widely available.

**Reviewer Confidence:**

4: Quite sure. I tried to check the important points carefully. It's unlikely, though conceivable, that I missed something that should affect my ratings.

**Typos Grammar Style And Presentation Improvements:**

- Line 84: use-s ?
- Line 191: Pys-Attention -> PayAttention
- Line 262+264: mutli_att -> multi_att ?
- Line 218+372: Missing space between Table and number (e.g., Table2 -> Table 2)
- Line 318: kaggle -> Kaggle
- whole paper: Either always "Table" n or always "table n" in a sentence (the same for Figure/figure)
- whole paper: either BigFive, Big Five or big five
- Figure 4: the caption refers to section 6.2 but the paper has only five sections
- Figure 5: the font size in the figure is too small
- Table 2 and 3 should also present the F1 measure
- Lines 350-369 should be part of Related Work"
- Section 4.6: The section does not really fit to the other content of the paper. Better use the space of this section to describe the attention mechanism and the Evaluation in more detail.
- Line 390-392: The claim that a difference of 0.3% "proves" that your model is "better" is brave (especially as the evaluation procedure is not described). You should interpret your results more realistic.
- Missing sections: Discussion of the methods, results and their limitations

---

> ### Author Rebuttal · Authors · 2023-08-26
>
> Thank you for your comments. We will correct the typos, add references to the missing literature, and revise the paper according to your suggestions.
> We will release our code when this paper is accepted.
>
>
> Question A: Which attention mechanism was used for the psychological feature encoder?
>
> Answer A: Self-attention
>
> Question B: How can any attention mechanism work on a sequence of discrete numerical values that do not represent meanings/semantics but just arbitrary values?
>
> Answer B:  We sincerely apologize for the oversight. In our modeling section, we unintentionally omitted a step. Before applying attention mechanisms to discrete numerical features, we utilize a fully connected neural network to generate a feature matrix for these discrete attributes. Then we use transformer’s encoder  to encode it. We deeply appreciate your question, and we will promptly rectify this error in our paper.
>
> Question C: How are the features represented when used as input for the encoder (this perhaps answers Question B too)?
>
> Answer C: We utilize a fully connected neural network to generate a feature matrix for these discrete attributes. These discrete attributes are obtained by three NLP tools such as SÉANCE, TAACO and TAALES.
>
> Question D: Can you describe how the datasets have been used for the evaluation (e.g., train/test/validation split, cross-validation (yes/no), distribution of labels in the dataset)?
>
> Answer D: We have employed a split ratio of 7:2:1 for training, validation, and testing respectively.   For the BigFive Essays dataset,  there are total 2468 items, for the label “O”, there are 1272 items are True and 1196 items are False, for the label “C”, there are 1254 items are True and 1214 items are False, for the label “E”, there are 1277 items are True and 1191 items are False,  for the label “A”, there are 1310 items are True and 1158 items are False,  for the label “N”, there are 1233 items are True and 1235 items are False.
>
> | BigFive Essays | O     | C     | E     | A     | N     |
> |-------|-------|-------|-------|-------|-------|
> | Ture  | 1272  | 1254  | 1277  | 1310  | 1233  |
> | False | 1196  | 1214  | 1191  | 1158  | 1235  |
> | Total | 2468  | 2468  | 2468  | 2468  | 2468  |
>
> For Kaggle MBTI dataset there are total 8675 items, for the label “I/E”, there are 6676 items are “I” and 1999 items are “E”, for the label “N/S”, there are 7478 items are “N” and 1197 items are “S”, for the label “T/F”, there are 3981 items are “T” and 4694 items are “F”,  for the label “P/J”, there are 3434 items are “P” and 5241 items are “J”.
>
> | Kaggle MBTI | I/E   | N/S   | T/F   | P/J   |
> |------------------|-------|-------|-------|-------|
> | I or N or T or P | 6676  | 7478  | 3981  | 3434  |
> | E or S or F or J | 1999  | 1197  | 4694  | 5241  |
> | Total            | 8675  | 8675  | 8675  | 8675  |
>
>
> Question E: As the balance of labels in the datasets not clear, what is the F1 measure of your results?
> Answer E:  We are sorry for it, as the baselines did not disclose their F1 scores, we did not report our F1 scores either.
> For MBTI the F1 values are as follows: “I/E” 0.8697, “N/S”0.8872,  “T/F” 0.8490   and “P/J” 0.7398
>
> Question F: Can motivate why it is useful to compare the text and feature embeddings by the cosine similarity?
>
> Answer F: We argue that the model extracts more psychological features from the text if the embedding of the text is more similar to the embedding of the psychological features, which allows the model to extract more psychological information contained in the text, thereby increasing the accuracy of the model.

---

### Official Review · Reviewer_y6Lk · 2023-07-26

**Soundness:** 3

**Excitement:**

4: Strong: This paper deepens the understanding of some phenomenon or lowers the barriers to an existing research direction.

**Missing References:**

BERT model is not cited in the paper.

**Paper Topic And Main Contributions:**

The paper identifies the use of more psychological features when fused with a deep learning model or used with a mere machine learning model as impeding respective models performance on personality detection, hence, the paper proposes a new model that devises an optimization approach to reduce the superfluous psychological features to a necessary size and feeds this to an attention-based encoder, the output of this is fused with BERT embeddings for personality detection.

**Questions For The Authors:**

A. Why addressing  BigFive and MBTI  are stated as models to be extended in Abstract and Introduction  sections while they are used as mere datasets in Experiments. It's better to just state them as datasets throughout the paper.
B. Why you didn't provide train/validation/test splits and statistics of data as used in the experiments.
C. Why presenting figure/table names as figure1/table and Figure 1/Table 1, for example. Why not to just use Figure 1 or Table 1?
D. Why not providing an elaborate explanation on the loss function?
E.  Where is 'HLayerNorm1' coming from in Equation 3?
F. Did you use pytorch, Tensorflow/keras, or what package for your model? How many epochs did you train your model? Did you use GPU, what type/name?


**Reasons To Accept:**

The paper clearly identifies the problem, proposes a novel way to address it and conducts experiments to support their claims and approach.

**Reasons To Reject:**

I only have these concerns from the paper:
1.  BigFive and MBTI  are stated as models to be extended in Abstract and Introduction  sections while they are used as mere datasets in Experiments. It's better to just state them as datasets throughout the paper unless the authors should provide an extended explanation why they are addressing them like that.
2. It's imperative to provide train/validation/test splits and statistics of data used in the experiments to aid in understanding the model performance, how the evaluation is being made, and for reproducibility.
3. Typos and presentation inconsistencies.

**Reproducibility:**

3: Could reproduce the results with some difficulty. The settings of parameters are underspecified or subjectively determined; the training/evaluation data are not widely available.

**Reviewer Confidence:**

5: Positive that my evaluation is correct. I read the paper very carefully and I am very familiar with related work.

**Typos Grammar Style And Presentation Improvements:**

Typos:       line
use-s         084
employe    532

Presentation Improvements:
On line 031, the sentence, "The content that users post on..." introduces a new idea I suggest to use a transition word/phrase or new paragraph.
On line 113, 'selected' should be 'select' since contributions are being proposed.
Be consistent in language use: On line 164, bidirectional LSTMs and BiLSTM are one thing hence first occurrence of them should be bidirectional LSTMs (BiLSTM) and use BiLSTM throughout the rest of the paper. Same for Natural Language Processing and NLP.  Where is 'HLayerNorm1' coming from in Equation 3?
On Table 1, I suggest MBTI and BigFive should be column headers and for 3rd and 4th columns  and "Number of Features after Optimization" should be used within table caption.
On Figure 1, label other components of model diagram in the middle upper area.
I suggest to use 'following' than 'follow' like sentence on line 305, "We can obtain the final embedding by follow formula".
I suggest to use 'pre-processing' than 'processing' on line 331.
Line 343 "head attention" should be attention head.
Line 372, put space between Table and 1 to be Table 1.
Restructure lines 370-376,  eg. add s in all occurrences of model since the term 'model' is representing plural things. and restructure as stated.
For Table 2, replace 'Essays' with 'BigFive'.
I recommend you should include  a table of ground-truth labels for all datasets.

---

> ### Author Rebuttal · Authors · 2023-08-26
>
> Thank you for your comments. We will correct the typos, add references to the missing literature, and revise the paper according to your suggestions.
>
> QA. Why addressing BigFive and MBTI are stated as models to be extended in Abstract and Introduction sections while they are used as mere datasets in Experiments. It's better to just state them as datasets throughout the paper.
>
> Answer: Thank you for comments, we apologize for the confusion, BigFive and MBTI are two different models used in psychology to measure an individual's personality, so we introduce them as models inside Abstract and Introduction. The names of the datasets in the experiment are actually BigFive Essays and Kaggle MBTI , for brevity we use BigFive and MBTI for this, we will revise this in future versions.
>
> QB. Why you didn't provide train/validation/test splits and statistics of data as used in the experiments.
>
> Answer: We apologize for the oversight in not providing an introduction to our train/validation/test splits. In actuality, we have employed a split ratio of 7:2:1 for training, validation, and testing respectively.  In the section of dataset, we give the statistics about the dataset, we are very sorry that we didn't make a table, we will add the statistics in a future version, thank you for your comments.
>
>
> QC. Why presenting figure/table names as figure1/table and Figure 1/Table 1, for example. Why not to just use Figure 1 or Table 1?
>
> Answer: Thank you for your suggestion, we will consistently use Figure 1 and Table 1 in future version.
>
>
> QD. Why not providing an elaborate explanation on the loss function?
>
> Answer: The final loss function of the model is the cross-entropy, and in section 3.3 Equation (6) is our own design of the loss function, Equation (6) is only for the Fine-tuning BERT part of our model.
>
> QE. Where is 'HLayerNorm1' coming from in Equation 3?
>
> Answer: We apologize for incorrectly entering "HLNorm1" as "HLayerNorm1", which is actually the result of Equation 2. Thank you for your comments.
>
> QF. Did you use pytorch, Tensorflow/keras, or what package for your model? How many epochs did you train your model? Did you use GPU, what type/name?
>
> Answer: We utilized Python 3.8.1, PyTorch 1.13.0, Transformers 4.24.0, and scikit-learn 1.1.3 for implementing our model. Our training process involved 4 NVIDIA RTX 3090 GPUs. We trained our model with 50 epochs.

---

### Official Review · Reviewer_imtg · 2023-08-03

**Soundness:** 4

**Excitement:**

3: Ambivalent: It has merits (e.g., it reports state-of-the-art results, the idea is nice), but there are key weaknesses (e.g., it describes incremental work), and it can significantly benefit from another round of revision. However, I won't object to accepting it if my co-reviewers champion it.

**Missing References:**

In the related work, you can mention this paper who deals with the problem of modelling personality rather than predicting it.
https://arxiv.org/abs/2301.08606


**Paper Topic And Main Contributions:**

The paper deals with the problem of psychological traits prediction. The traits are taken from psychological theory (the Big Five model for example). The paper proposes a new approach to fine-tuning BERT, incorporating psychological features extracted using existing pipelines, into the training phase (in the loss function). A test is performed on two large datasets and about 5% improvement in accuracy is obtained. Also, the method proposes a way to perform feature selection and reduce the number of psychological variables that are extracted using existing pipelines (which is in the hundreds).

**Questions For The Authors:**

When you report accuracy, adding the chance accuracy (the prevalence of that label) is better.

**Reasons To Accept:**

The paper is well-written, the method is explained well, and seems innovative.
The improvement is significant.
The problem is central in NLP

**Reasons To Reject:**

The problem of identifying psychological features was extensively studied, and while improvement is offered by the new method, all in all the result is relatively incremental.

**Reproducibility:**

4: Could mostly reproduce the results, but there may be some variation because of sample variance or minor variations in their interpretation of the protocol or method.

**Reviewer Confidence:**

2: Willing to defend my evaluation, but it is fairly likely that I missed some details, didn't understand some central points, or can't be sure about the novelty of the work.

---

> ### Author Rebuttal · Authors · 2023-08-26
>
> Thank you for comments, we will incorporate "the chance accuracy" in the next version, as well as the referenced paper you recommended.

---

### Meta-Review · Area_Chair_njqw · 2023-09-04

**Recommendation:** 3
**Confidence:** 5

**Metareview:**

The problem of identifying psychological features was extensively studied, and while improvement is offered by the new method, all in all the result is relatively incremental. During review period, the paper under review has obtained substantial positive feedback, particularly for its high-quality writing and well-articulated methodology. It investigate into a central problem in Natural Language Processing (NLP), offering an innovative approach to predicting an author's personality based on their text. The authors excel in giving a comprehensive overview of methods for extracting personality-related features from text, highlighting the paper's significance in advancing the field.

While the paper is largely commendable, there are areas identified for further refinement, which have minor impacts on its overall quality. For instance, the paper could benefit from greater detail to improve its reproducibility. Including specifics about the attention mechanism on numerical sequences and the input representation for the psychological feature encoder would make the work even more robust.
Reviewer 2's insights, though critical, offer constructive avenues for strengthening the paper. The presentation of BigFive and MBTI models could be clarified by consistently referring to them as datasets, unless a more elaborate explanation is provided. Additionally, the inclusion of details on train/validation/test splits and data statistics could significantly enhance understanding and replicability, while also addressing minor issues like typos and presentation inconsistencies.

Lastly, while it's noted that the research problem has been explored in prior works, the paper still makes a worthwhile, albeit incremental, contribution to the existing literature. The novelty and quality of the paper indicate that with a few refinements, it could serve as an impactful piece in the NLP community. Overall I would like to see this paper into the poster sessions.

---

### Decision · Program_Chairs · 2023-10-07

**Decision:**

Accept-Findings

**Comment:**

The problem of identifying psychological features was extensively studied, and while improvement is offered by the new method, all in all the result is relatively incremental. During review period, the paper under review has obtained substantial positive feedback, particularly for its high-quality writing and well-articulated methodology. It investigate into a central problem in Natural Language Processing (NLP), offering an innovative approach to predicting an author's personality based on their text. The authors excel in giving a comprehensive overview of methods for extracting personality-related features from text, highlighting the paper's significance in advancing the field.

While the paper is largely commendable, there are areas identified for further refinement, which have minor impacts on its overall quality. For instance, the paper could benefit from greater detail to improve its reproducibility. Including specifics about the attention mechanism on numerical sequences and the input representation for the psychological feature encoder would make the work even more robust.
Reviewer 2's insights, though critical, offer constructive avenues for strengthening the paper. The presentation of BigFive and MBTI models could be clarified by consistently referring to them as datasets, unless a more elaborate explanation is provided. Additionally, the inclusion of details on train/validation/test splits and data statistics could significantly enhance understanding and replicability, while also addressing minor issues like typos and presentation inconsistencies.

Lastly, while it's noted that the research problem has been explored in prior works, the paper still makes a worthwhile, albeit incremental, contribution to the existing literature. The novelty and quality of the paper indicate that with a few refinements, it could serve as an impactful piece in the NLP community. Overall I would like to see this paper into the poster sessions.